# ANCA Associated Glomerulonephritis Following SARS-CoV-2 Vaccination: A Case Series and Systematic Review

**DOI:** 10.3390/vaccines11050983

**Published:** 2023-05-15

**Authors:** Theerachai Thammathiwat, Athiphat Banjongjit, Kroonpong Iampenkhae, Natavudh Townamchai, Talerngsak Kanjanabuch

**Affiliations:** 1Division of Nephrology, Department of Medicine, Faculty of Medicine, Naresuan University, Phitsanulok 65000, Thailand; 2Nephrology Unit, Department of Medicine, Vichaiyut Hospital, Bangkok 10400, Thailand; 3Department of Pathology, Faculty of Medicine, Chulalongkorn University, Bangkok 10330, Thailand; 4Division of Nephrology, Department of Medicine, Faculty of Medicine, Chulalongkorn University and King Chulalongkorn Memorial Hospital, Bangkok 10330, Thailand; 5Excellence Center for Solid Organ Transplantation, King Chulalongkorn Memorial Hospital, Bangkok 10330, Thailand; 6Renal Immunology and Renal Transplant Research Unit, Chulalongkorn University, Bangkok 10330, Thailand; 7Center of Excellence in Kidney Metabolic Disorders, Faculty of Medicine, Chulalongkorn University, Bangkok 10330, Thailand

**Keywords:** SARS-CoV-2 vaccination, COVID-19 vaccine, ANCA-associated pauci-immune glomerulonephritis, ANCA-associated vasculitis

## Abstract

Vaccines against SARS-CoV-2 (COVID-19) proved beneficial for COVID-19 disease attenuation and preventing virus spreading. Cumulative reports of the rarity of antineutrophil cytoplasmic autoantibodies (ANCA)-associated vasculitis (AAV) raise concerns about its relationship with COVID-19 vaccination. Several case reports described ANCA-associated pauci-immune glomerulonephritis (ANCA-GN) following COVID-19 vaccination with some uniqueness. We systematically reviewed COVID-19 vaccine-induced ANCA-GN from PubMed, SCOPUS, and Cochrane library databases until 1 January 2023 according to PRISMA guidelines and presented our three cases. Twenty-six cases from 25 articles, including our 3 cases, were analyzed. Most cases were diagnosed following the second dose of the COVID-19 vaccine (59%) with a median (IQR) interval onset of 14 (16) days. The highest prevalence was related to the mRNA-type vaccine. Anti-myeloperoxidase (MPO) ANCA was far more common than the other ANCAs, with various positive autoantibodies. Fourteen cases (out of 29 cases, 48%) had extra-kidney AAV manifestation. Although severe kidney injury was observed in 10/29 (34%), remission was achieved in 89% (25/28) with no death. The mechanisms of the vaccine-inducing ANCA-GN were postulated here. Since ANCA-GN after the COVID-19 vaccine was rare, the benefit of the COVID-19 vaccine could outweigh the risk of ANCA-GN side effects in the pandemic era.

## 1. Introduction

The severe acute respiratory syndrome coronavirus 2 (SARS-CoV-2) caused the ongoing global pandemic of coronavirus disease 2019 (COVID-19), with a cumulative confirmed case of 765 million, including a 1% death rate [1]. Most COVID-19 vaccines are designed to elicit immune responses, ideally neutralizing antibodies against the SARS-CoV-2 spike protein, including mRNA (BNT162b2 and mRNA-1273), adenoviral-vectored (AZD1222, Ad26.COV2.S, and Gam-COVID-Vac), protein subunit (NVX-CoV2373), and whole-cell inactivated virus vaccines (CoronaVac, Sinopharm, BBIBP-CorV, and BBV152) [2]. COVID-19 vaccines prove their efficacy profiles on COVID-19 prevention and reduction in disease severity according to a recent meta-analysis of randomized controlled trials [3]. In addition, COVID-19 vaccine safety profiles were acceptable without increased risk of serious adverse events [3]. However, several de-novo and relapsing glomerulonephritis (GN) following COVID-19 vaccination, including minimal change disease (MCD), IgA nephropathy (IgAN), membranous nephropathy (MN), lupus nephritis (LN), anti-glomerular basement membrane (anti-GBM) disease, and ANCA-associated pauci-immune glomerulonephritis (ANCA-GN) have been documented [4,5,6]. ANCA-GN is a severe glomerular disease manifesting as systemic vasculitis involving glomeruli from ANCA-associated vasculitis (AAV) [7,8,9,10]. A recent systematic review of AAV following COVID-19 vaccines demonstrated a temporal relationship [11]. However, renal manifestation and outcomes of biopsy-proven ANCA-GN following the COVID-19 vaccine were limited. In this review, we presented our 3 cases and systematically reviewed biopsy-proven ANCA-GN following COVID-19 vaccines from the literature. We also discussed the disease’s uniqueness, possible mechanism, treatment, and outcomes.

## 2. Materials and Methods

### 2.1. Eligibility Criteria 

The American College of Rheumatology/European Alliance of Associations for Rheumatology (ACR/EULAR) 2022 classified AAV into granulomatous polyangiitis (GPA), eosinophilic granulomatous polyangiitis (EGPA), microscopic polyangiitis (MPA), and renal-limited vasculitis (RLV) [12,13]. However, this is only a classification criterion for research purposes [12]. Therefore, in the systematic review, we included only kidney-biopsy-proven ANCA-GN following COVID-19 vaccination with or without extra-kidney manifestation of AAV. 

### 2.2. Search Strategy and Study Selection

Two independent authors conducted systematic literature by searching relevant full-text articles in the PubMed, SCOPUS, and Cochrane library databases up to 1 January 2023. Keywords were (“COVID-19 vaccine”, OR “COVID-19”, OR “COVID-19 vaccination”, OR “SARS-CoV-2 vaccine”, OR “SARS-CoV-2”) AND (“ANCA”, OR “ANCA-associated glomerulonephritis”, OR “ANCA-associated vasculitis”, OR “glomerulonephritis”, OR “MPO-ANCA”, OR “PR3-ANCA”, OR “pauci-immune glomerulonephritis”, OR “anti-neutrophil cytoplasmic antibody”, OR “antineutrophil cytoplasmic antibody”). The eligible articles included the full-text report in the English language of biopsy-proven ANCA-GN with pauci-immune deposit by immunofluorescence (IF) study (defined as negative or weak [≤1+] staining of Ig and complement) or by electron microscopy (EM) (defined as no or faint electron-dense deposit [EDDs]) after COVID-19 vaccination. Patients’ age, vaccine types, and the disease onset from the vaccination were not restricted. The exclusion criteria were AAV without GN, ANCA-GN with a non-available IF/EM report, and AAV presenting or having disease flare before the onset of vaccination. We also included our 3 GN-ANCA cases diagnosed and treated at King Chulalongkorn Memorial Hospital, Bangkok, Thailand, from January 2022 to December 2022. Data extraction was anonymized under the General Data Protection Regulation [14], and informed consent, per the Declaration of Helsinki, were obtained from all three patients or their descendants (if unavailable). The PRISMA flow is shown in Figure 1.

### 2.3. Data Extraction

Data included patient demographics, vaccine types, vaccination numbers, the interval between the last vaccination and onset of AAV, the clinical manifestation of ANCA-GN and AAV, serum creatinine (Scr) at baseline and peak, serologies (e.g., cytoplasmic [c] ANCA, perinuclear [p] ANCA, MPO-ANCA, PR3-ANCA, anti-nuclear antibody [ANA], Coombs antibodies, cryoglobulin, rheumatoid factor [RF]), kidney pathology finding, treatment, and outcomes after treatment.

### 2.4. Data Analysis and Definition

Severe acute kidney injury (AKI) was defined as dialysis requirement or peak Scr > 5.7 mg/dL. Outcomes were classified into remission with relapse, remission without relapse, and non-remission (refractory, dialysis-dependent, and death). Remission of ANCA-GN/AAV was defined according to the 2012 KDIGO Guidelines for Management of Glomerular Diseases [15] as the absence of erythrocyte in the urine, stable or reduced presence of proteins in the urine, and stable or improved glomerular filtration rate (GFR), together with the absence of symptoms indicating active disease in any extra-kidney organ system. 

The continuous data were illustrated by using mean ± standard deviation (SD) for normal distribution data and median (interquartile range [IQR]) for non-normal distribution data. The categorical data were described as a ratio. Finally, the normally distributed variables were compared by *t*-test. 

## 3. Results

### 3.1. Systematic Review

A total of 2956 relevant records were initially obtained from the database search. After removing 312 duplicated records, 2644 record titles and abstracts of articles were first screened according to the inclusion criteria. 68 articles were eligible for the second screening; 43 articles (154 cases) were excluded (4 with irrelevant to inclusion criteria, 5 no patients detail, 10 review articles, 20 without kidney-biopsy/IF confirmation; and 4 with negative ANCA); 26 cases from 25 articles, including our 3 cases were recruited for the systematic review (Figure 1). The details of the individual cases are shown in Appendix A. 

The median (IQR) age was 70 (22) years, with female predominance (59%). Most cases presented ANCA-GN after receiving the second vaccination (59%) with a median interval gap of 14 (11–27) days. The mRNA COVID-19 vaccine was the most common (55%), followed by the viral vector vaccine (31%) and inactivated COVID-19 vaccine (14%). The top-three most common presentations were AKI, abnormal urine sediments, and constitutional symptoms. Extra-kidney manifestations included pulmonary involvement (diffuse alveolar hemorrhage and interstitial pneumonia) [8/29, 28%], neuromuscular involvement (muscle weakness, arthralgia/arthritis) [8/29, 28%], otologic/optic involvement [5/29, 17%], and Wallerian degeneration [1/29, 3%]. Severe AKI was observed in 34% (10/29). MPO-ANCA/pANCA was positive in 79% (23/29). A discrepancy of ANCA serologies was observed in one case (1/29 cases, cANCA + MPO-ANCA). A dual-positive ANCA (pANCA and cANCA) was detected in 3% (1/29). Autoantibodies were positive in 52% (13/25 cases), including ANA (64%, 7/11), Coombs antibody (100%, 3/3), cryoglobulin (33%, 1/3), and RF (25%, 2/8). The 2021 KDIGO Guidelines for Management of Glomerular Diseases [16] were adopted to treat most cases with ANCA-GN following COVID-19 vaccines. Plasmapheresis was prescribed in 21% (6/28), while rituximab was employed in 38% (11/29). Remission was achieved in 89% (25/28), with relapse in one of those. In non-remission, 11% (3/28) were dialysis-dependent, and no death occurred (Table 1).

### 3.2. Case Presentations

We described three cases of ANCA-GN following viral vectors (case #1) and mRNA (case #2 and #3) COVID-19 vaccines. Only one patient (case #2) was classified as microscopic polyangiitis (MPA) by ACR/EULAR 2022 [17]. 


*Case #1*


A 76-year-old female with type 2 diabetes mellitus, hypertension, and dyslipidemia presented with leg edema for 11 days prior to admission. She received the second COVID-19 vaccine (AZD1222) 3 months ago. A few weeks later, she developed chronic fever, weight loss, and productive cough. Several blood cultures were performed, yielding negative results. Non-contrast chest computed tomography (CT) revealed unusual interstitial pneumonia (UIP). Sputum cultures depicted normal flora. She denied a history of SARS-CoV-2 infection and had negative PCR for COVID-19. Her Scr was 1.5 mg/dL over the past year. Three-week before admission, her Scr increased to 1.7 mg/dL with 2+ proteinuria and erythrocyte of 10–20/HPF. A week later, she developed leg edema and foamy urine. Physical examination (PE) revealed temperature 38.9 °C, respiratory rate (RR) 20/min, blood pressure (BP) 165/91 mmHg, and bilateral 3+ leg pitting edema. Blood chemistries revealed hemoglobin 7.6 g/L, leukocyte count of 11.9 × 10^9^/L with a neutrophil predominance (83%), Scr 3.5 mg/dL, and albumin 2.5 g/dL. Urinalysis demonstrated similar profiles with dysmorphic erythrocytes. The urine protein-creatinine ratio (UPCR) was 7.5 g/g creatinine. pANCA was positive with negative MPO-ANCA and PR3-ANCA. Serologies were positive for ANA (speckled pattern, 1:1280), RF 30 (0–20) IU/mL, and Coombs antibody (1+), while others were negative (e.g., anti-GBM antibody, viral hepatitis profiles, treponemal, and anti-HIV antibody). Complement levels were within normal values. The kidney biopsy disclosed 24 glomeruli with 7 global sclerosis (GS) and 7 cellular crescents. Fibrinoid necrosis of arteries and severe tubulointerstitial inflammation were observed. Interstitial fibrosis and tubular atrophy involved 30% of cortical tissue. IF revealed IgG (1+), C3 (1+), Kappa (1+), and Lambda (1+). Focal trivial subepithelial and intramembranous EDDs with diffuse thickening of GBM were observed on EM. Hence, ANCA-GN without extra-kidney organ involvement was diagnosed. Three-day of 1 gm/day pulse methylprednisolone and 500 mg pulse cyclophosphamide was started. Unfortunately, her kidney function remained deteriorated, requiring maintenance hemodialysis.


*Case #2*


A 69-year-old female presented with AKI while investigating anorexia, weight loss, and fatigue. She had hypertension for 2 decades and received 2 doses of AZD1222 and a COVID-19 vaccine booster with BNT162b2 (3 months ago). One month later, she developed anorexia, fatigue, a weight loss of 8 kg within 2 months, and established mixed conductive and sensorineural hearing loss. Chest X-ray and contrast computerized tomography (CT) revealed a subcentimeter pulmonary nodule. Two weeks after CT, her Scr increased from 0.7 to 2.4 mg/dL, and contrast-associated AKI was suspected. However, despite receiving an intravenous fluid infusion, she had a steadily increasing Scr. Her urinalysis revealed 1+ protein, leukocyte 50–100/HPF, and erythrocyte 3–5/HPF. Ciprofloxacin was prescribed on suspicion of having a urinary tract infection. PE revealed temperature 38.0 °C, BP 120/60 mmHg, marked pale conjunctivae, and no leg edema. Blood chemistries were notable for Scr 7.1 mg/dL, albumin 2.5 g/dL, hemoglobin 6.6 g/dL, leukocyte counts 12.1 × 10^9^/L with neutrophils predominance (83%), and platelet counts 386 × 10^9^/L. Urinalysis depicted trace protein, erythrocytes 3–5 cells/HPF, and UPCR 0.5 g/g creatinine. pANCA and MPO-ANCA were positive, with negative PR3-ANCA and cANCA. Serologies were positive, including ANA (speckled pattern, 1:80), cryoglobulin, and 1+ direct antiglobulin test, while the others were negative (e.g., anti-GBM antibody, RF, viral hepatitis profiles, treponemal, and anti-HIV antibody). Complement levels were within normal values. Serum protein electrophoresis and immunofixation demonstrated polyclonal gammopathy. No structural kidney abnormalities were detected by ultrasonography. The kidney biopsy disclosed 13 glomeruli with 2 GS and 10 cellular crescents. Fibrinoid necrosis was found on the vasculitic medium-sized artery. Mild tubulointerstitial fibrosis was observed. IF revealed non-specific staining for all immunoreactants. Hence, AAV with ANCA-GN was diagnosed. She received plasma exchange ×7 times, pulse methylprednisolone, and oral cyclophosphamide. One year later, she remained well without dialysis requirements and active AAV manifestation. Her serum creatinine 2.3 mg/dL and negative erythrocyturia.


*Case #3*


An 84-year-old female with underlying diseases of hypertension and dyslipidemia presented with a fever for one week prior to admission. She received complete primary series of AZD1222 and a booster dose of the COVID-19 vaccine with mRNA-1273 four months ago. One month later, she developed bilaterally severe sensorineural hearing loss and tinnitus. A few weeks before admission, she developed drowsiness, progressive dysphagia, decreased urine volume, and a weight loss of 8 kg. PE revealed BP 154/66 mmHg, temperature 38.5 °C, mildly pale conjunctivae, right eye episcleritis with afferent pupillary defect (RAPD), and 1+ leg edema. Blood chemistries were notable for Scr 5.1 mg/dL (baseline 0.9 mg/dL), albumin 2.7 g/dL, hemoglobin 9.6 g/dL, leukocyte counts 26.0 × 10^9^/L with neutrophils predominance (80%), and platelet counts 209 × 10^9^/L. Urinalysis depicted 2+ protein, leukocytes 30–50/HPF, erythrocytes 1–2/HPF, and UPCR 1.2 g/g creatinine. cANCA and MPO-ANCA were positive, with negative PR3-ANCA. Positive direct antiglobulin test at 1+ was detected, while others were negative (e.g., anti-GBM antibody, ANA, RF, viral hepatitis profiles, treponemal, and anti-HIV antibody). Complement levels were normal. Serum protein electrophoresis and immunofixation demonstrated polyclonal gammopathy. No structural kidney abnormalities were detected by ultrasonography. The kidney biopsy disclosed 20 glomeruli with 2 GS and 7 cellular crescents. Fibrinoid necrosis was noted in 6 glomeruli. Diffuse tubulointerstitial inflammation with abundant eosinophils and trivial tubulointerstitial fibrosis (<5%) were observed. IF revealed non-specific staining for all immunoreactants. ANCA-GN with eosinophilic interstitial nephritis was diagnosed. Due to the alteration of consciousness, magnetic resonance imaging (MRI) of the brain and orbits were performed, demonstrating Wallerian degeneration of the right corticospinal tracts and pons and bilateral optic neuritis. Intravenous pulse methylprednisolone and intravenous immunoglobulin were administered as per AAV. Temporary hemodialysis was initiated for 2 weeks. Subsequently, she remained well without AAV clinical flare with low-dose prednisolone and immunosuppressive agents; her Scr was 1.6 mg/dL with negative proteinuria and bland urine sediment.

## 4. Discussion

ANCA-GN secondary to COVID-19 vaccination was more prevalent in the patients with mRNA vaccinations (a median gap interval of 14 days), older age, and female gender. The top three most common manifestations were AKI, abnormal urine sediment, and constitutional symptoms. MPO-ANCA was distinctly dominant compared to the other ANCAs. Concomitant autoantibodies were documented in 52%, particularly Coombs antibody, ANA, and cryoglobulin. Although one-third (34%) presented with severe ANCA-GN, most cases had good outcomes, with a remission rate of 89%. No death occurred. 

The higher immunogenicity and more common usage of the mRNA COVID-19 vaccine might explain its higher prevalence on ANCA-GN [18,19]. The median onset of ANCA-GN following COVID-19 vaccination was much shorter than other drug-induced ANCA-GN (14 days vs. 9 months) [20]. The difference might be related to the booster effect of the vaccine since most presented cases followed the second and third COVID-19 vaccinations. The booster dose of the COVID-19 vaccine increased the absolute risk of glomerular disease compared to naïve exposure [21]. The supporting evidence was noted in thioridazine-induced ANCA-AAV (leukocytoclastic vasculitis) that occurred 4 weeks after first exposure compared to a shorter duration of 24 h after rechallenging the dose [22] similar to minocycline-induced cutaneous lupus 1–2 days after repeated exposure compared to 23 months for the first exposure [23]. Moreover, one case series reported early onset of ANCA-AAV (12 days to 3 weeks) post-influenza vaccination [24].

Most AAVs have unknown etiology, known as primary AAVs; however, AAVs can be secondary to drugs (e.g., propylthiouracil, hydralazine, levamisole-contaminated cocaine) [9,25,26,27], vaccination (e.g., influenza and rabies vaccines) [28,29], and hematologic malignancies (e.g., leukemias, lymphomas) [26]. SARS-CoV-2 can infect host cells using the angiotensin-converting enzyme 2 (ACE2) receptor, expressed predominantly on endothelial cells’ surface, causing endotheliitis and hypercoagulable stage [30]. However, the vascular effects of COVID-19 vaccination have yet to be known. Irure-Ventura et al. [31] found that the incidence of new ANCA-positive patients had increased in 2021 compared to 2019 (2.4% vs. 1.2%).

AAV/ANCA-GN induced by vaccination has been well documented in post-influenza vaccination [28,32]. The authors performed an *in-vitro* experiment and found that the influenza vaccine (RNA vaccine) was able to induce PR3-ANCA production from peripheral blood mononuclear cells (PBMCs), while the decoy molecule, inactivated split virion influenza vaccine 2007, could not [33]. In addition, the AAV manifestations responded well to ribonuclease treatment in the vaccinated patients [33,34]. Unfortunately, the pathogenesis of ANCA-GN followed by the COVID-19 vaccine remains obscure [35]. 

COVID-19 vaccine induces ANCA-GN and AAV, possibly through molecular mimicry, stimulating productions of MPO-ANCA and PR3-ANCA via the adaptive immune system, likewise inducing antiviral neutralizing antibodies [36,37,38]. Then the ANCA ignites the inflammation cascade pathway by producing inflammatory cytokines (e.g., type 1 interferon, interleukin-6), activating Toll-like receptor (TLR), priming and activating neutrophils, causing neutrophils extracellular traps (NETs) formation, a mesh-like structure comprising DNA, histone, and neutrophil granules (e.g., MPO, PR3). Thus, MPO and PR3 on the NETs might further fuel the inflammatory fire and viciously start the ANCA production cascade [9,35,36,37,39]. mRNA vaccination in mice, monkeys, and humans activates TLR3, TLR4, and TLR7, causing robust cytotoxic T-lymphocyte activation [40,41,42]. Another possible mechanism of vaccine-inducing AVV involves monocytes upregulating human leukocyte antigen (HLA)-DR, which encodes cell-surface proteins responsible for regulating the immune system, in the patient at risk of AAV, having HLA-DR4 or DRB4 alleles [11,43]. Some HLA-DR alleles are connected with autoimmune disorders [43]. Our review found that concomitant positive autoantibodies were common (more than half) in COVID-19 vaccine-induced ANCA-GN/AAV patients. However, the interplaying mechanism between COVID-19 vaccine-induced ANCA-GN and autoantibodies needs further exploration. 

Treatment of ANCA-GN following the COVID-19 vaccines in this systematic review mostly complied with the 2021 KDIGO Guidelines [16], whereby using a combination of glucocorticoid and cyclophosphamide or rituximab as the treatment backbone. Plasmapheresis was employed only in cases with severe AKI, double-positive ANCA, and anti-GBM antibody, or pulmonary hemorrhage [13,16]. Outcomes of ANCA-GN following COVID-19 were good, albeit severe manifestation, paradoxical to primary ANCA-GN with a remission rate of 89% compared to 39–48% [44]. However, this study has some limitations. First, the study included only articles in the English language. Secondly, this study pooled observational studies; thus, claiming a cause-effect relationship should be cautious. 

## 5. Conclusions

ANCA-associated glomerulonephritis following the COVID-19 vaccine is rare, according to billions of vaccine doses that have been administered worldwide. The benefits of COVID-19 vaccination outweigh the risks of possible induced autoimmune diseases and ACNA-GN. Avoid unnecessary booster doses of the COVID-19 vaccine in high-risk patients (carrying HLA-DR4 or DRB4 alleles) should warrant.

## Figures and Tables

**Figure 1 vaccines-11-00983-f001:**
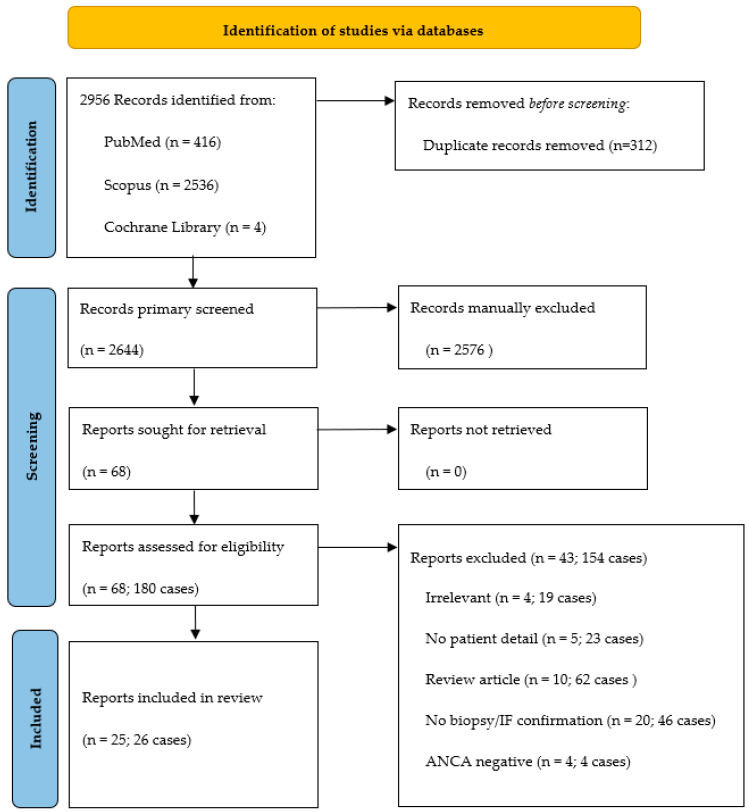
PRISMA flow diagram.

**Table 1 vaccines-11-00983-t001:** Clinical spectrums and outcomes of ANCA-associated glomerulonephritis following SARS-CoV-2 vaccination by vaccine type.

Parameters	Total (*n* = 29)	mRNA Vaccine (*n* = 16)	Viral Vector Vaccine (*n* = 9)	Inactivated Vaccine (*n* = 4)	*p* Value
Age, years	70 (22)	70 (25)	72 (17)	69 (33)	0.63
Female	17/29 (59%)	9/16 (56%)	5/9 (56%)	3/4 (75%)	0.88
Vaccination dose					0.08
● 1st dose	9/29 (31%)	3/16 (19%)	5/9 (56%)	1/4 (25%)	
● 2nd dose	17/29 (59%)	12/16 (75%)	2/9 (22%)	3/4 (75%)	
● 3rd dose or more	3/29 (10%)	1/16 (6%)	2/9 (22%)	0/4 (0%)	
Onset after last vaccination, days	14 (16)	14 (24)	15 (8)	20 (16)	0.72
Clinical manifestations					
● AKI or RPGN	14/29 (48%)	8/16 (50%)	5/9 (56%)	1/4 (25%)	0.76
● Sub-nephrotic proteinuria/NS	9/29 (31%)	3/16 (19%)	4/9 (44%)	2/4 (50%)	0.30
● Abnormal urine sediments	19/29 (66%)	10/16 (62%)	5/9 (56%)	4/4 (100%)	0.40
● Constitutional symptoms	15/29 (52%)	6/16 (38%)	5/9 (56%)	4/4 (100%)	0.10
● Extra-kidney involvement					
- Lung (DAH, interstitial pneumonia, etc.)	8/29 (28%)	5/16 (31%)	3/9 (33%)	0/4 (0%)	0.62
- NM (muscle weakness, etc.)	8/29 (28%)	3/16 (19%)	4/9 (44%)	1/4 (25%)	0.40
- GI (N/V, diarrhea, etc.)	4/29 (14%)	1/16 (6%)	1/9 (11%)	2/4 (50%)	0.10
- ENT (hearing loss, tinnitus, etc.)	3/29 (10%)	1/16 (6%)	2/9 (22%)	0/4 (0%)	0.70
- Eyes (episcleritis, etc.)	2/29 (7%)	1/16 (6%)	1/9 (11%)	0/4 (0%)	1.00
ANCA serologies					
● MPO-ANCA/pANCA	23/29 (79%)	13/16 (81%)	7/9 (78%)	3/4 (75%)	1.00
● PR3-ANCA/cANCA	5/29 (17%)	3/16 (19%)	1/9 (11%)	1/4 (25%)	1.00
● pANCA + cANCA	1/29 (3%)	0/16 (0%)	1/5 (8%)	0/4 (0%)	1.00
● Discrepancy positive ANCA	1/29 (3%)	0/16 (0%)	1/6 (17%)	0/4 (0%)	1.00
Autoantibodies, n/N (%)					
● ANA	7/11 (64%)	5/5 (100%)	1/4 (20%)	1/2 (50%)	0.05
● Coombs/positive anti-globulin test	3/3 (100%)	1/1 (100%)	2/2 (200%)	-	-
● Cryoglobulin	1/3 (33%)	1/3 (33%)	-	-	-
● RF	2/8 (25%)	0/4 (0%)	2/3 (67%)	0/1 (0%)	0.21
Baseline Scr, mg/dL	0.9 (0.3)	0.9 (0.2)	1.0 (0.7)	0.8 (0.7)	0.14
Peak Scr, mg/dL	4.8 (3.5)	3.5 (3.7)	4.8 (2.7)	5.9 (2.6)	0.60
De-novo GN	26/29 (90%)	15/16 (94%)	8/9 (89%)	4/4 (100%)	1.00
Treatment					
● Corticosteroid	29/29 (100%)	16/16 (100%)	9/9 (100%)	4/4 (100%)	-
● Rituximab	11/29 (38%)	9/16 (56%)	2/9 (22%)	0/4 (0%)	0.09
● Plasmapheresis	6/28 (21%)	6/15 (40%)	0/9 (0%)	0/4 (0%)	0.04
● Hemodialysis	8/29 (28%)	2/16 (12%)	4/9 (44%)	2/4 (50%)	0.14
Follow-up time, weeks	8 (8)	8 (7)	14 (10)	24 (40)	0.48
Outcomes					0.65
● Remission with relapse	1/28 (3%)	1/16 (6%)	0/8 (0%)	0/4 (0%)	
● Remission without relapse	24/28 (86%)	14/16 (88%)	6/8 (75%)	4/4 (100%)	
● Non-remission	3/28 (11%)	1/16 (6%)	2/8 (25%)	0/4 (0%)	

**Remarks:** Continuous variables were presented with median and interquartile range, while categorical variables were presented with frequency and percentage. **Abbreviations:** AKI, acute kidney injury; ANCA, antineutrophil cytoplasm antibody; ANA, anti-nuclear antibodies; cANCA, cytoplasmic ANCA; DAH, diffuse alveolar hemorrhage; ENT, ear nose throat; GI, gastrointestinal; GN, glomerulonephritis; MPO, anti-myeloperoxidase; NM, neuromuscular; N/V, nausea/vomiting; PR3, anti-proteinase 3; pANCA, perinuclear ANCA; RPGN; rapidly progressive glomerulonephritis; RF, rheumatoid factor.

## Data Availability

Not applicable.

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
