# Peer review of "ANCA Associated Glomerulonephritis Following SARS-CoV-2 Vaccination: A Case Series and Systematic Review"

_vaccines, 2023, doi:10.3390/vaccines11050983_

Round 1
Reviewer 1 Report
The manuscript by Thammathiwat et al a case A Case Series and Systematic Review of ANCA associated glomerulonephritis following SARs-CoV2 2 vaccination. This systemic review is informative and provide evolving issues with covid 19 vaccines. The review is a systematic search of literature of relevant articles with inclusion of 3 case reports. Since there are previous reports of systematic review related to ANCA-GN with COVID-19 vaccine the current review needs significant justification and explanation for publication.
Author Response
Reviewer #1
The manuscript by Thammathiwat et al a case A Case Series and Systematic Review of ANCA associated glomerulonephritis following SARs-CoV2 2 vaccination. This systemic review is informative and provide evolving issues with covid 19 vaccines. The review is a systematic search of literature of relevant articles with inclusion of 3 case reports.
- Since there are previous reports of systematic review related to ANCA-GN with COVID-19 vaccine the current review needs significant justification and explanation for publication.
Response: The previous systematic review conducted by Baier et al. entitled "Dual-Positive MPO- and PR3-ANCA-Associated Vasculitis Following SARS-CoV-2 mRNA Booster Vaccination: A Case Report and Systematic Review." included only de-novo AAV following COVID-19 vaccination, while our systematic review specifically included ANCA-GN with or without AAV. Pathological criteria were employed to specify the diagnosis of ANCA-GN. In addition, treatment outcomes were included in our review while not in the previous study. Furthermore, we have grouped the patients' manifestation, serologies, and treatment outcomes accordingly to the type of COVID-19 vaccine.

Reviewer 2 Report
The English language of the manuscript needs to be thoroughly improved throughout the paper.Lines 46-48: Check this sentence - it does not sound correct. For example, the word vaccine in line 47 seems to be misplaced.
Lines 53-54: This sentence is too vague and not informative. Provide more details, correct it. Introduction: In general, this section lacks the main aspect - providing background on the issue and outlining the research done so-far and stating the aim. Line 70: There is no description of the facility where the cases were treated. The time/date when the case was treated is not provided. Line 212: Why were only two databases searched? Methods: Provide a clear list of inclusion and exclusion criteria. Lines 402-405: The limitations are not clearly stated. Here it must be stated that only papers published in the English language were considered. Further on, what does the sentence on lines 402-403 mean? Existence of previous studies is a limitation?Author Response
Reviewer #2
- The English language of the manuscript needs to be thoroughly improved throughout the paper.
Response: English language in the whole manuscript has been revised accordingly.
- Lines 46-48: Check this sentence - it does not sound correct. For example, the word vaccine in line 47 seems to be misplaced.
Response: Please accept our apology for the mistakes. We have edited the text accordingly.
“Since ANCA-GN after the COVID-19 vaccine was rare, the benefit of the COVID-19 vaccine could outweigh the risk of ANCA-GN side effects in the pandemic era.”
- Lines 53-54: This sentence is too vague and not informative. Provide more details, correct it.
Response: The sentence has been revised accordingly, as below.
“SARS-CoV-2 (COVID-19) vaccines prove their efficacy with acceptable safety profiles on COVID-19 prevention and reduction in disease severity according to the results of a recent meta-analysis of randomized controlled trials [1]. Several de-novo and relapsing glomerulonephritis (GN) following COVID-19 vaccination, including minimal change disease (MCD), IgA nephropathy (IgAN), membranous nephropathy (MN), lupus nephritis (LN), anti-glomerular basement membrane disease (anti-GBM), and ANCA-associated pauci-immune glomerulonephritis (ANCA-GN) have been documented [2-4]. ANCA-GN is a severe glomerular disease manifesting as systemic vasculitis involving glomeruli from ANCA-associated vasculitis (AAV) [5-8].”
- Introduction: In general, this section lacks the main aspect - providing background on the issue and outlining the research done so-far and stating the aim.
Response: Thank you for your valuable comment. The introduction section has been revised accordingly as below.
“SARS-CoV-2 (COVID-19) vaccines prove their efficacy with acceptable safety profiles on COVID-19 prevention and reduction in disease severity according to the results of a recent meta-analysis of randomized controlled trials [1]. Several de-novo and relapsing glomerulonephritis (GN) following COVID-19 vaccination, including minimal change disease (MCD), IgA nephropathy (IgAN), membranous nephropathy (MN), lupus nephritis (LN), anti-glomerular basement membrane disease (anti-GBM), and ANCA-associated pauci-immune glomerulonephritis (ANCA-GN) have been documented [2-4]. ANCA-GN is a severe glomerular disease manifesting as systemic vasculitis involving glomeruli from ANCA-associated vasculitis (AAV) [5-8]. A recent systematic review of AAV following COVID-19 vaccines demonstrated a temporal relationship [9]. However, renal manifestation and outcomes of biopsy-proven ANCA-GN following the COVID-19 vaccine were limited. In this review, we presented our 3 cases and systematically reviewed biopsy-proven ANCA-GN following COVID-19 vaccines from the literature. We also discussed the disease's uniqueness, possible mechanism, treatment, and outcomes.”
- Line 70: There is no description of the facility where the cases were treated. The time/date when the case was treated is not provided.
Response: The materials and methods section has been revised accordingly, as below.
“We also included our 3 GN-ANCA cases diagnosed and treated at King Chulalongkorn Memorial Hospital, Bangkok, Thailand, from January 2022 to December 2022. Data extraction was obtained with appropriate permission and consent by local guidelines.”
- Line 212: Why were only two databases searched?
Response: The Cochrane Library database has been added to our systematic review accordingly, as shown in Figure 1.
- Methods: Provide a clear list of inclusion and exclusion criteria.
Response: The materials and methods section has been revised accordingly.
"The eligible articles included the full-text report in the English language of biopsy-proven ANCA-GN with pauci-immune deposit by immunofluorescence (IF) study (defined as negative or weak [≤1+] staining of Ig and complement) or by electron microscopy [EM] (defined as no electron-dense deposit [EDDs]) after COVID-19 vaccination. Patients' age was not restricted. The exclusion criteria were AAV without GN, ANCA-GN with a non-available IF/EM report, and AAV presenting or having disease flare before the onset of vaccination.”
- Lines 402-405: The limitations are not clearly stated. Here it must be stated that only papers published in the English language were considered. Further on, what does the sentence on lines 402-403 mean? Existence of previous studies is a limitation?
Response: The limitations have been added and revised to clarify our messages as below.
“However, this study has some limitations. First, the study included only articles in the English language. Secondly, this study pooled observational studies; thus, claiming a cause-effect relationship should be cautious.”

Reviewer 3 Report
1- Please review the language as there are some minor errors.
2- Figure 2 should be part of the results section rather than the methods section.
3- Under the results section for the systematic review, you need to describe the results presented in figure 1 including the reasons for excluding some papers.
4- I believe that the structure of the manuscript should change. The case reports should go under the results section and the methods section should include a section for the case reports.
5- I also suggest that you use the table as a supplementary material and to create 3 smaller tables for the 3 vaccines and to cluster the data in each table based on the symptoms. Those tables should be smaller and focus mainly on the clinical aspects while the supplementary table will be the comprehensive source of data.
6- The new tables will make it easier to interpret the results and to follow the discussion section.
Author Response
Reviewer #3
- Please review the language as there are some minor errors.
Response: The English language has been edited accordingly.
- Figure 1 should be part of the results section rather than the methods section.
Response: Per your suggestion, we have moved Figure 1 out of the methods section.
- Under the results section for the systematic review, you need to describe the results presented in figure 1, including the reasons for excluding some papers.
Response: The results section has been revised accordingly.
"A total of 2,956 relevant records were initially obtained from the database search. After removing 312 duplicated records, 2,644 record titles and abstracts of articles were first screened according to the inclusion criteria. 68 articles were eligible for the second screening; 43 articles (154 cases) were excluded (4 with irrelevant to inclusion criteria, 5 no patients detail, 10 review articles, 20 without kidney-biopsy/IF confirmation; 4 with negative ANCA); 26 cases from 25 articles, including our 3 cases were recruited for the systematic review. (Figure 1).”
- I believe that the structure of the manuscript should change. The case reports should go under the results section and the methods section should include a section for the case reports.
Response: The materials & methods and results sections have been revised accordingly.
- I also suggest that you use the table as a supplementary material and to create 3 smaller tables for the 3 vaccines and to cluster the data in each table based on the symptoms. Those tables should be smaller and focus mainly on the clinical aspects while the supplementary table will be the comprehensive source of data. The new tables will make it easier to interpret the results and to follow the discussion section.
Response: Table 1 has been revised, and the supplementary table has been added accordingly.

Reviewer 4 Report
The manuscript describes important novel data in the studied area and worth to be published with some improvements in the presentation style and language. Discussion section shall be expanded via additional perspectives regarding immunohematological basis of COVID-19 vaccine AE field (such as; Hematological aspects of the COVID-19 syndrome. by Malkan UY, et al. Eur Rev Med Pharmacol Sci. 2022 Jun;26(12):4463-4476. doi: 10.26355/eurrev_202206_29086.PMID: 35776048)
Author Response
Reviewer #4
The manuscript describes important novel data in the studied area and worth to be published with some improvements in the presentation style and language.
- Discussion section shall be expanded via additional perspectives regarding immunohematological basis of COVID-19 vaccine AE field (such as; Hematological aspects of the COVID-19 syndrome. by Malkan UY, et al. Eur Rev Med Pharmacol Sci. 2022 Jun;26(12):4463-4476. doi: 10.26355/eurrev_202206_29086.PMID: 35776048)
Response: The recommended article has been cited in the discussion section accordingly.
“SARS-CoV-2 can infect host cells using the angiotensin-converting enzyme 2 (ACE2) receptor, expressed predominantly on endothelial cells' surface, causing endotheliitis and hypercoagulable stage [27].”.

Round 2
Reviewer 1 Report
The authors addressed the reviewers comments adequately.
Author Response
Thank you for your valuable comments and suggestions.Reviewer 2 Report
I appreciate the Authors' effort to revise the manuscript according to the received comments. Most of the issues were addressed.
The section Introduction was revised, but it still lacks the main parts. Namely, the section should start off with a brief one or two sentences that present an overview of the pandemic and developed vaccines, followed by a note or two on the safety profile of vaccines. Inclusion and exclusion criteria: Were there any criteria regarding the vaccine types, time from vaccination etc.? Lines 79-80: Cite these guidelines. Define "appropriate permission", add a reference number for the received approval. The Limitations have been revised and now they present well the study's characteristics.Author Response
Reviewer #2
I appreciate the Authors' effort to revise the manuscript according to the received comments. Most of the issues were addressed.
- The section Introduction was revised, but it still lacks the main parts. Namely, the section should start off with a brief one or two sentences that present an overview of the pandemic and developed vaccines, followed by a note or two on the safety profile of vaccines.
Response: The Introduction section has been revised accordingly, as below.
“The severe acute respiratory syndrome coronavirus 2 (SARS-CoV-2) caused the ongoing global pandemic of coronavirus disease 2019 (COVID-19), with a cumulative confirmed case of 765 million, including a 1% death rate [1]. Most COVID-19 vaccines are designed to elicit immune responses, ideally neutralizing antibodies (Abs) against the SARS-CoV-2 spike protein, including mRNA, adenoviral-vectored, protein subunit, and whole-cell inactivated virus vaccines [2].”
"In addition, COVID-19 vaccine safety profiles were acceptable without increased risk of serious adverse events [3]."
- Inclusion and exclusion criteria: Were there any criteria regarding the vaccine types, time from vaccination etc.?
Response: Vaccine types and the disease onset from vaccination were not restricted.
- Lines 79-80: Cite these guidelines. Define "appropriate permission", add a reference number for the received approval.
Response: The Materials and Methods section has been revised accordingly, as below.
"Data extraction was anonymized under the General Data Protection Regulation [14], and informed consents, per the Declaration of Helsinki, were obtained from all three patients or their descendants (if unavailable)."
- The Limitations have been revised and now they present well the study's characteristics.
Response: Thank you for your review and valuable comment.
Reviewer 3 Report
The manuscript has much improved.
Author Response
Reviewer #3
- The manuscript has much improved.
Response: Thank you for your valuable comments and suggestions.